# Mediterranean Diet versus Very Low-Calorie Ketogenic Diet: Effects of Reaching 5% Body Weight Loss on Body Composition in Subjects with Overweight and with Obesity—A Cohort Study

**DOI:** 10.3390/ijerph192013040

**Published:** 2022-10-11

**Authors:** Claudia Di Rosa, Greta Lattanzi, Chiara Spiezia, Elena Imperia, Sara Piccirilli, Ivan Beato, Gianluigi Gaspa, Vanessa Micheli, Federica De Joannon, Noemi Vallecorsa, Massimo Ciccozzi, Giuseppe Defeudis, Silvia Manfrini, Yeganeh Manon Khazrai

**Affiliations:** 1Unit of Food Science and Human Nutrition, Campus Bio-Medico University of Rome, 00128 Rome, Italy; 2Unit of Gastroenterology, Department of Medicine, University Campus Bio-Medico of Rome, 00128 Rome, Italy; 3Unit of Medical Statistic and Molecular Epidemiology, Department of Medicine, University Campus Bio-Medico of Rome, 00128 Rome, Italy; 4Unit of Endocrinology and Diabetes, Department of Medicine, University Campus Bio-Medico of Rome, 00128 Rome, Italy

**Keywords:** ketogenic diet, Mediterranean diet, 5% weight loss

## Abstract

The best nutritional strategy to fight the rise in obesity remains a debated issue. The Mediterranean diet (MD) and the Very Low-Calorie Ketogenic diet (VLCKD) are effective at helping people lose body weight (BW) and fat mass (FM) while preserving fat-free mass (FFM). This study aimed to evaluate the time these two diets took to reach a loss of 5% of the initial BW and how body composition was affected. We randomized 268 subjects with obesity or overweight in two arms, MD and VLCKD, for a maximum of 3 months or until they reached 5% BW loss. This result was achieved after one month of VLCKD and 3 months of MD. Both diets were effective in terms of BW (*p* < 0.0001) and FM loss (*p* < 0.0001), but the MD reached a higher reduction in both waist circumference (*p* = 0.0010) and FM (*p* = 0.0006) and a greater increase in total body water (*p* = 0.0017) and FFM (*p* = 0.0373) than VLCKD. The population was also stratified according to gender, age, and BMI. These two nutritional protocols are both effective in improving anthropometrical parameters and body composition, but they take different time spans to reach the goal. Therefore, professionals should evaluate which is the most suitable according to each patient’s health status.

## 1. Introduction

Obesity is a multifactorial disease that deteriorates the quality of life, and it is now considered a growing pandemic that imposes a heavy burden on families and society [1]. It is a chronic condition defined as an abnormal or excessive fat accumulation that may impair health [2]. It is a risk factor for multiple comorbidities (i.e., type 2 diabetes (T2DM), hypertension, cardiovascular diseases (CVD), cancer, and sleep apnea), and it is associated with increased mortality risk [3]. Currently more than 1.9 billion adults are overweight, of which 650 million suffer from obesity [2]. In Italy, 4 out of 10 adults are overweight, and 1 out of them is affected by obesity [4]. Overweight and obesity are classified based on the Body Mass Index (BMI). A BMI range between 25.0 and 29.9 defines overweight, while a BMI > 30 determines obesity [5]. Achieving a healthy weight is considered a risk modifier and has favorable effects on blood pressure, glucose metabolism, and cardiac and vascular functions [1]. In the literature, it has been demonstrated that even 5% of body weight loss may improve health outcomes, and this value has been set as a goal standard in weight loss interventions [6,7]. Many patients with obesity [8,9] and type 2 diabetes [10] are not aware that losing at least 5% of their body weight could improve their quality of life. Indeed, losing at least 5% of body weight creates a significant reduction in fat mass (8 ± 3%) and not only an improvement in multi-organ insulin sensitivity and beta cell function but also in blood pressure and heart rate, as was seen in a recent RCT on 40 subjects with obesity [7]. This modest reduction has also shown beneficial effects on CVD risk factors at 1 year in patients with obesity and type 2 diabetes. In fact, the loss of 5% body weight created a significant reduction in diastolic and systolic blood pressure and glycemic and lipid metabolism (except for LDL cholesterol, which did not change significantly) [10]. Moreover, these studies showed that losing more than 5% provided further health benefits [7,10].

Sometimes the time necessary to reach the goal of losing 5% of body weight is a key factor, since people with overweight or with obesity may lose their motivation and abandon the nutritional protocol, starting the so-called yo-yo effect [11]. It is important to establish a feasible objective reachable through lifestyle changes and healthy eating habits [9]. Generally, these two behavioral changes require time to be adopted, and when choosing the most suitable dietary pattern, it is important to consider not only the general health benefits but also the individual [6,9].

The Mediterranean Diet (MD) and Very Low-Calorie Ketogenic Diet (VLCKD) are often recommended by health professionals for their beneficial effects on weight loss (especially fat mass, FM) and the preservation of fat-free mass (FFM) [12,13,14,15,16].

In 2010, the Mediterranean Diet (MD) became an intangible heritage of humanity by UNESCO [17]. This nutritional approach is considered sustainable and provides a high consumption of local fruit and vegetables, whole grains, legumes, nuts and seeds, blue fish, eggs, white meats, and dairy products, and a low consumption of red and processed meats with respect to correct portion sizes and weekly frequencies [18,19]. MD also supports correct hydration, extra virgin olive oil as a seasoning, home-made food preparation, sociable eating with family or friends, regular physical activity, relaxation, and rest [20,21]. The MD suggests consuming more low-energy-density foods [22], and compared with many other dietary patterns (i.e., the Western diet), it provides a low glycemic load because it is rich in foods high in dietary fibers, which give great satiety after a meal [23]. A hypocaloric MD showed greater weight loss than the low-fat diet at 12 and 24 months but similar weight loss compared to other diets (i.e., low-carbohydrate diets and calorie-restricted diets) [24]. Epidemiological studies have noticed increased longevity and reduced morbidity in Mediterranean countries, which follow the MD, compared to the USA or Northern Europe populations [25,26]. The MD not only helps with weight loss maintenance but also has beneficial effects on cardiovascular risk factors, cognitive functions, and mood [27,28].

On the other hand, a ketogenic diet is characterized by no more than 30–50 g/day of carbohydrates [29,30]. This carbohydrates deprivation depletes glycogen stores; thus, the body undergoes metabolic changes to provide an energy source for the body and the brain through gluconeogenesis and ketogenesis. Through this last process, ketone bodies (KB) (acetoacetate, beta-hydroxybutyrate, and acetone) are produced, and they are used as the primary energy source by cells with mitochondria and by all organs, particularly the brain [16]. This ketosis state, called “physiological ketosis”, should not be confused with diabetic ketoacidosis [31] because in the first case, ketonemia reaches maximum levels of 7–8 mmol/L with no changes in pH (7.4), whilst in the second case, ketonemia can exceed 20 mmol/L with a concomitant lowering of blood pH (<7.3) [32]. At present, there are different types of ketogenic diets, with or without calorie restriction: Isocaloric Ketogenic Diet (IKD), Low-Calorie Ketogenic Diet (LCKD), and the Very-Low-Calorie Ketogenic Diet (VLCKD) [12]. The VLCKD provides less than 700–800 kcal/day with carbohydrates ranging between 30 and 50 g per day, preferably <30 g, and <30–40 g/day of fats, mainly from extra virgin olive oil [12,13,32]. It is a normal protein diet; in fact, it should provide high values of biological proteins in the amount of 0.8–1.5 g/kg ideal body weight/day to preserve free fat mass [12,32,33]. The VLCKD must be supplemented with bicarbonate, micronutrients, and omega-3 fatty acids, and it should only be followed for short periods (8–16 weeks) [12,32]. A period of VLCKD may help control hunger and improve fat-oxidative metabolism, and therefore reduce body weight, but the transition from a VLCKD to a standard diet should be gradual and well-controlled [16]. In the literature, it has been seen that a VLCKD creates a significant reduction in body weight, BMI, waist circumference, and fat mass (especially visceral fat) while preserving lean mass [34,35,36,37]. These body composition changes seem to be associated with better food control and quality of life, which could contribute to maintaining weight loss over time [36]. Results obtained after a VLCKD seem to be better than results obtained with other nutritional protocols (very low-calorie diets, low-calorie diets, low-fat diets) in the same time span [36]. The Italian Society of Endocrinology recommends the VLCKD as an effective dietary treatment for people with obesity, hypertriglyceridemia, obesity associated with T2DM [14], hypertension [13,32], particularly for severe obesity and/or comorbidities (joint diseases, preoperative period of bariatric surgery, and cardiovascular and metabolic diseases) who need a rapid and substantial weight loss. The VLCKD can be prescribed to a specific population of patients after considering the potential contraindications and keeping the patients under medical surveillance. The VLCKD is contraindicated for people affected by type 1 diabetes mellitus (T1DM), recent cardiovascular or cerebrovascular events, severe hepatic and renal insufficiency, gout episodes, kidney stones, hydroelectrolytic alterations, psychiatric disease, and pregnant and breastfeeding women [15,32].

The present study aimed to observe how long a MD and a VLCKD took to create a loss of at least 5% of the initial body weight in subjects with overweight and obesity and how these two nutritional protocols affect anthropometrical measures and body composition parameters.

## 2. Materials and Methods

### 2.1. Study Population

Three hundred and seventy-four (374) males and females subjects with overweight or obesity were consecutively enrolled in the Endocrinology and Metabolic Diseases unit of the University Hospital Campus Bio-Medico of Rome from December 2021 to May 2022 according to the inclusion and exclusion criteria. Inclusion criteria were adults aged 18–70 years and BMI ≥ 25 kg/m^2^. Exclusion criteria comprised subjects with Type 1 Diabetes Mellitus (T1DM), T2DM, pregnancy and breastfeeding, kidney failure and severe chronic kidney disease, liver failure, hearth failure (NYHA III–IV), respiratory insufficiency, unstable angina, a recent stroke or myocardial infarction (<12 months), cardiac arrhythmias, eating disorders and other severe mental illnesses, alcohol and substance abuse, active/severe infections, and frail older adults. Consecutive enrolment was repeated in the two groups, either a Mediterranean diet (MD) or a Very Low-Calorie Ketogenic Diet (VLCKD), achieving casualty in allocation, and thus randomization from December 2021 to May 2022. The MD group comprised 191 subjects (54 males, 137 females), while the VLCKD group comprised 183 subjects (46 males, 137 females). The two groups were homogeneous for age, height, and weight.

To evaluate the effects of these two different nutritional protocols, the population was stratified according to gender (male and female), age (≤50 years old or ≥50 years old), and BMI (in subjects with overweight BMI comprised between 25–29.9 kg/m^2^; in subjects with obesity, BMI > 30 kg/m^2^).

### 2.2. Study Protocol

At baseline (T0), the subjects underwent an endocrinological visit to evaluate if they could be included in the study according to the exclusion and inclusion criteria. When enrolled, they were consecutively allocated to either the MD or VLCKD. Participants received nutritional counseling and were given the dietetic protocol to be followed for three months. Follow-up visits were set up once monthly until 5% body weight loss was achieved.

Every fortnight, a phone call was conducted by nutritionists to check the dietary adherence and patients’ motivation through a 24 h dietary recall (Figure 1). When patients reached the goal of losing at least 5% of their initial body weight, they were invited to stop following the nutritional protocol and to start a maintenance diet (for the MD group) or a reintroduction diet (for the VLCKD group).

### 2.3. Anthropometric Parameters

At T0, body weight (BW) and height were recorded using a Seca 200 scale with a stadiometer. Measurements were taken in the morning after an overnight fast of at least 12 h. Subjects wore underwear and were without shoes. Height was measured only at T0, while body weight was recorded every month until participants lost at least 5% of their initial body weight for a maximum of three months for both diets.

BMI was calculated as weight (kg)/height (m^2^) [5] and waist circumference (WC) was measured in the medium point between the last rib and the iliac crest [38].

### 2.4. Body Composition Analysis

Body composition was assessed using bioelectrical impedance analysis (BIA, Akern 101 New Edition). BIA measurements were performed at T0 and once a month until the 5% weight loss for a maximum of three months. Measurements were conducted in the morning after an overnight fast of at least 12 h, in abstinence of alcohol consumption for 48 h and without strenuous physical activity for 24 h before the testing day. The parameters obtained by this exam were: phase angle, total body water (TBW), extracellular water (ECW), intracellular water (ICW), fat-free mass (FFM), fat mass (FM), and body cellular mass (BCM) [39].

### 2.5. Nutritional Protocols

#### 2.5.1. Mediterranean Diet

The hypocaloric Mediterranean dietary plan was individualized according to participants’ nutritional needs and preferences. Each participant’s total energy expenditure (TEE) was calculated, and then a caloric restriction of 500 kcal was applied. On average, the diet provided 1500 kcal for women and 1700 kcal for men. The macronutrient composition of the diet was 15% protein, 30–35% lipids, and 50–55% carbohydrates, with less than 15% comprising simple sugars. The diet included five meals daily (breakfast, lunch, dinner, and two snacks) both for men and women. Participants were asked to prefer vegetables, wholegrain cereals, fish and legumes, lean white meat, and seeds, and to reduce red meat, eggs, and dairy products to only once a week. They were invited to consume fruit or low-fat yogurt as snacks and extra virgin olive oil as a seasoning in the amount of 30 mL (for women)–40 mL (for men) daily. Women were recommended to drink at least 2 L of water, while men were instructed to drink at least 2.5 L daily [40].

#### 2.5.2. Very Low-Calorie Ketogenic Diet

The VLCKD provided <800 kcal and <30–50 g of carbohydrate/day. The amount of protein was calculated as 1.2–1.5 g of protein/kg ideal body weight/day, respectively, for women and men. The diet was composed of 4 daily meals for women and 5 daily meals for men, consisting of both meal replacements and natural foods. The diet was structured as follows: meal replacements for breakfast and snacks and white or red meat, fish, eggs, smoked salmon, ham, or canned fish with a vegetable side dish for lunch and dinner. Subjects were allowed to use extra virgin olive oil as a seasoning in the amount of 30 g/day, both for men and women. A standard sugar-free multivitamin, bicarbonate, minerals (potassium 2000 mg and magnesium 375 mg), and omega 3 fatty acids (1 g), were prescribed. Drinking at least 2–2.5 L of water was recommended for women and for men, respectively [40].

### 2.6. Statistical Analysis

Statistical analysis was performed with the statistical program GraphPad Prism version 9.4.0. A Pearson test was conducted to evaluate the normality distribution. Paired and unpaired student *t*-tests were performed for intragroup and intergroup comparisons, respectively. The significance level was assessed for *p*-value <0.05.

The sample size of 128 subjects (64 for each group) was calculated with an 80% power to detect changes at the 0.05 significance level, taking possible dropouts into account.

## 3. Results

### 3.1. Study Population

Out of 374 subjects, only 268 completed the study. The 106 dropouts missed either the 1 month visit or the 2-month visit without losing at least 5% of their initial body weight after the first month.

The group which followed the MD until the end of the study was composed of 133 subjects (37 males and 96 females; mean age 45.08 ± 14.19 years), while the group that followed the VLCKD comprised 135 subjects (34 males and 101 females; mean age 45.5 ± 11.63 years) (Figure 2).

### 3.2. Results on Anthropometric Parameters and Body Composition

VLCKD subjects lost at least 5% of their initial body weight after one month of the dietary regime, while the MD group reached this goal after 3 months. In the VLCKD group the percentage of body weight lost was 7.21 ± 1.57% after one month, while in the MD group the percentage was 7.68 ± 2.52% after three months. Both groups lost at least 5% of their initial body weight, but in two different time spans. There were no significant differences in body weight loss between the two groups.

The results of both diets on anthropometric and body composition parameters are reported in Table 1. Comparing the results obtained from MD and VLCKD, we observed a higher reduction in waist circumference (−6.86 ± 3.3 cm vs. −5.74 ± 2.07 cm; *p* = 0.0010) and fat mass percentage (−3.15 ± 2.49 vs. −2.17 ± 2.14; *p* = 0.0006) in the MD group compared to the VLCKD group. Moreover, the Mediterranean diet provided a greater increase of total body water percentage (2.27 ± 1.96% vs. 1.58 ± 1.61; *p* = 0.0017) and fat-free mass percentage (2.89 ± 3.08 vs. 2.23 ± 1.99; *p* = 0.0373) than the Very Low-Calorie Ketogenic diet.

We stratified MD and VLCKD groups according to gender (male or female) to observe the effects of these two nutritional protocols on anthropometric and body composition parameters. There were 37 males (mean age: 45.22 ± 16.81 years) and 96 females (mean age: 45.03 ± 13.14 years) in the MD group, while in the VLCKD group, there were 34 males (mean age: 44.56 ± 13.25 years) and 101 females (mean age: 45.81 ± 11.09 years).

The results of both diets on anthropometrical and body composition parameters in males and females are reported in Table 1.

Comparing all parameters between MD and VLCKD in the male group, we did not observe any significant differences. In relation to females, we observed that the MD provided better results than the VLCKD in the reduction in waist circumference (−6.8 ± 3.5 cm vs. −5.54 ± 1.97 cm; *p* = 0.0020) and fat mass percentage (−3.27 ± 2.69% vs. −2.24 ± 2.21%; *p* = 0.0034). Moreover, a higher increase in total body water percentage (2.42 ± 2.09 vs. 1.72 ± 1.49; *p* = 0.0076) and fat-free mass percentage (3.06 ± 2.73% vs. 2.31 ± 1.92%; *p* = 0.0251) was observed in the MD group compared to the VLCKD group.

### 3.3. Subgroups Evaluation: Age and BMI

#### 3.3.1. Age

We also evaluated the two diets according to age (≤50 years old or ≥50 years old) to observe if, in older adults, there could be a greater reduction in fat-free mass and body cellular mass, increasing the risk of sarcopenia.

In the MD group there were 78 subjects (23 males and 55 females) ≤ 50 years old (mean age 35.5 ± 9.50 years) and 55 subjects (14 males and 41 females) ≥ 50 years old (mean age 58.6 ± 6.80 years), while in the VLCKD group, there were 78 subjects (20 males and 58 females) ≤ 50 years old (mean age 38.17 ± 9.67 years) and 57 (14 males and 43 females) ≥ 50 years old (mean age 55.52 ± 4.26 years).

Observing the effects of the two nutritional protocols in people under 50, the MD was more efficient than the VLCKD in the reduction in waist circumference (6.96 ± 3.26 cm vs. 5.75 ± 2.2 cm, respectively; *p* = 0.0076), total body water percentage (2.39 ± 2.06% vs. 1.54 ± 1.52%, respectively; *p* = 0.0040), fat-free mass percentage (3.12 ± 2.76% vs. 2.31 ± 1.79%, respectively; *p* = 0.0326), and increase in fat mass percentage (3.33 ± 2.64% vs. 2.15 ± 1.97%, respectively; *p* = 0.0019). No significant differences were seen comparing the results between the two over 50 groups. We did not observe any changes in the body cellular mass percentage in both groups under and over 50 years old, so we can assume that these two diets preserve the body cellular mass at every age.

The results of both diets on anthropometrical and body composition parameters in subjects under and over 50 years old are reported in Table 2.

#### 3.3.2. BMI

Finally, we divided the population according to BMI: subjects with overweight (BMI comprised between 25.0–29.9 kg/m^2^) and with obesity (BMI > 30 kg/m^2^).

In the MD group, we had 46 subjects with overweight (37 females and 9 males; mean BMI: 27.55 ± 1.56 kg/m^2^) and 87 with obesity (59 females and 28 males; mean BMI: 34.57 ± 3.89 kg/m^2^), while in the VLCKD group we had 38 subjects with overweight (33 females and 5 males, mean age: 43.11 ± 13.41; mean BMI: 27.67 ± 1.37 kg/m^2^) and 97 with obesity (68 females and 29 males; mean BMI: 35.84 ± 4.72 kg/m^2^).

Comparing the two nutritional protocols in subjects with overweight, we observed that the MD was more efficient than the VLCKD in the reduction in waist circumference (6.64 ± 2.68 cm vs. 5.00 ± 1.81 cm, respectively; *p* = 0.0020) and fat mass percentage (3.84 ± 2.59% vs. 2.58 ± 1.72%, respectively; *p* = 0.0126). The MD also caused a higher increase in total body water percentage (2.79 ± 2.02 with MD and 1.99 ± 1.19 with VLCKD; *p* = 0.0340). There were no significant differences in fat-free mass percentage between the two nutritional protocols.

Among subjects with obesity, we observed that the MD determined a higher reduction than the VLCKD in terms of waist circumference (6.98 ± 3.59 cm vs. 6.03 ± 2.11 cm, respectively; *p* = 0.0289) and of fat mass percentage (2.79 ± 2.38% vs. 2.01 ± 2.27%, respectively; *p* = 0.0237). No significant differences were observed for the fat-free mass percentage between the two diets. Regarding water balance, the Mediterranean diet caused a higher increase compared to the Very Low-Calorie Ketogenic diet’s total body water percentage (2.00 ± 1.88% vs. of 1.42 ± 1.73%, respectively; *p* = 0.0301). The extracellular water percentage decreased significantly in the VLCKD group compared to the MD group (*p* = 0.0409).

The results of both diets on anthropometrical and body composition parameters in subjects with overweight or with obesity are reported in Table 3.

## 4. Discussion

The reduction by at least 5% of one’s body weight is linked to improved health outcomes and quality of life for people with overweight and obesity [9]. The European Guidelines for obesity management in adults set this modest weight loss as a goal for people affected by overweight or obesity, underlining that the more weight lost, the more beneficial health effects are seen [41], as also confirmed in other studies [7,10].

In the literature, few studies have evaluated the effects of reaching 5% body weight loss in terms of anthropometric parameters and body composition [7]. Our study focused on observing the time necessary to reach this target at different ages and BMIs by comparing two nutritional protocols on anthropometric and body composition parameters.

The results of this study confirmed those of other ones in the literature where both the MD and VLCKD were efficient in reducing body weight, waist circumference, and fat mass while preserving FFM and BCM in subjects with overweight or obesity [24,34,35,36,37,42,43]. The goal to lose at least 5% of one’s body weight was reached after a month of VLCKD and after 3 months of MD. In the study of Magkos et al., after 3.5 months, participants lost 5% of their initial body weight following a hypocaloric diet, and they decreased FM by 8%± 3% but also had a 2%± 2% reduction in FFM [7]. This result is not in line with ours, because in our study we did not observe an FFM reduction. The preservation of FFM, especially of body cellular mass, is an important issue, as its reduction alters energy metabolism and decreases muscle strength as well as lung and immune capacity [44,45]. Maximizing fat loss while preserving lean mass and its function is a central goal of overweight and obesity treatment [46]; in fact, fat-free mass represents a key determinant of the magnitude of one’s resting metabolic rate (RMR) [47].

As in other studies in the literature, we observed a significant reduction in body weight, BMI, WC, and FM after both diets in both male and female groups. The comparison between male groups of both diets showed no significant differences. A couple of literature studies that evaluated the effects of a VLCKD on the male population were focused on its effect on testicular function [48] or metabolic hypogonadism and beta cell function [49]. Both studies were longer than ours (at least 12 weeks) and had a different aim, so they did not evaluate body composition changes, only body weight and BMI as anthropometrical parameters. In any case, even in these studies, the researchers observed a significant reduction in body weight and BMI compared to baseline [48,49]. Moreover, the effects of MD on anthropometric and body composition parameters in a male population could be seen in the study of Carneiro-Barrera et al., in which 75 males were randomized to a usual care group or an eight-week weight loss and lifestyle intervention group to observe an improvement in sleep apnea. They reached their goal after a significant reduction in body weight (*p* < 0.0001) and fat mass (*p* < 0.0001) thanks to a Mediterranean diet [50]. In terms of female results, in our MD group, there was both a greater reduction in waist circumference and fat mass percentage and a higher increase in total body water percentage and fat-free mass percentage compared to the VLCKD group. Our study confirmed other results in the literature about weight loss, BMI, and WC reduction after one month of VLCKD, and they are also in line with FFM preservation [51]. The study of Barrea et al. observed a significant increase in the phase angle (considered an inflammation marker) in 260 women after 1 month of VLCKD, but in our study, we did not observe any changes in the phase angle in all subgroups [52]. Similar results on anthropometric parameters in women with overweight or obesity were seen in the study of Tragni et al. [53]. This study used 24 weeks of the VLCKD because it also accounted for the reintroduction phase. In the whole study, patients reduced body weight (−14.6%) and waist circumference (−12.4%), and at the end of the protocol, 33% of participants reached a normal weight [53]. We observed similar results on anthropometric parameters, but our study was shorter because it only considered the active phase.

To evaluate the effects of these two different nutritional protocols on FFM, we stratified the population according to age. We chose the threshold of 50 years old because generally, after the 50th year, significant aging processes take place, and every year after the 50th there could be a small physiological loss of fat-free mass and strength [54]. Even if muscle mass tends to decrease physiologically with age, we did not observe any reductions in fat-free mass and body cellular mass after both dietary regimes. Although a diet poor in carbohydrates may increase muscle catabolism [55], exacerbating fat-free mass loss during weight loss, the VLCKD preserved FFM and body cellular mass. The same result was seen after the MD. Indeed, both dietary plans could be considered safe for the risk of sarcopenia. We noticed that in subjects younger than 50 years, the MD was more effective than the VLCKD in reducing waist circumference (*p* = 0.0076) and fat mass percentage (*p* = 0.0019) and in increasing free-fat mass percentage (*p* = 0.0326) and total body water percentage (*p* = 0.0040). In the >50 group, there was no difference between the MD and VLCKD, but we noticed that intragroup, there was a significant increase in FFM (*p* < 0.0001 for both diets), and BCM was not affected by weight loss. Other studies in the literature did not divide by age, and they usually observed the results on the whole population, which comprised both younger and elder subjects.

## 5. Conclusions

The original aspect of this study was evaluating the time necessary to at least achieve the goal of 5% of body weight loss with two different dietary treatments. We observed that this result was achieved through one month of the Very-Low-Calorie Ketogenic diet and three months of the Mediterranean diet.

These two nutritional protocols are adequate for both men and women of different age groups with overweight or obesity. Both dietary programs induced weight and fat mass loss without affecting free-fat mass and body cellular mass.

It is clear that the MD is a nutritional protocol useful for the prevention and management of non-communicable diseases such as obesity and metabolic and cardiovascular diseases with high diet adherence and satisfaction by patients [56], but nowadays, the Very-Low-Calorie Ketogenic diet has become even more popular, especially due its rapid effect on weight loss. Thus, people, especially those with severe obesity, are more motivated to follow this nutritional protocol to attain faster results. However, it is important to remember that a VLCKD is not sustainable in the long term, and it requires a gradual transition to a Mediterranean diet [57]. Therefore, combining these two nutritional therapies might be a winning strategy to help people healthily lose weight, increasing their motivation [11,29,32,33].

The results of this study are promising. The large number of participants and the small number of dropouts are surely a strength of the study. Amongst limitations of the study there is that we did not monitor levels of physical activity and diet adherence with a food diary but relied on participants’ reports of their food intakes during the fortnight phone calls and follow-up visits. This could determine a bias, as subjects might forget some details about their food and drink intakes, as recently highlighted in a systematic review. People tend to frequently misestimate their food portions and sometimes forget their consumptions of some food, such as vegetables or seasonings [58]. In the present study, patients’ reports about their food intakes, in terms of portion sizes and frequencies, were in accordance with those established by the study protocol. Moreover, in our study we focused only on anthropometrical and body composition changes rather than on biochemical assessment (i.e., glycemia and lipid profile). The European Guidelines for obesity management in adults underline that VLCKDs determine greater reduction of total cholesterol and serum triglycerides but do not improve glycemic levels, HbA1c, LDL cholesterol, or HDL cholesterol compared to other nutritional protocols for the same time span [14]. For a future study, it could be interesting to evaluate the effects of the two nutritional patterns considered in the study on glycemic and lipid profile once the goal of 5% of body weight loss is reached. Another limitation of this study is that our results refer to a short period of time, while it would be interesting to know if these improvements are maintained over time. Studies in the literature show that after 6 months a VLCKD determines more significant results compared to a hypocaloric diet on weight loss, but after 12 months this difference is no more significant. In fact, people who follow a hypocaloric diet are able to continue losing weight over time, while people who follow a VLCKD tend to regain a bit of weight during or after the reintroduction phase [59]. Our study, in fact, showed that in two different time spans the results on body weight and body composition are similar, but it could be interesting to schedule a follow-up visit after 6 or 12 months to check weight maintenance, as this is the most challenging aspect of diet therapy [60]. To avoid weight gain after a diet period, it is important to define realistic goals to gradually change lifestyle habits and maintain weight loss over time [41]. These lifestyle changes could be obtained through nutrition education and the permanent acquisition of healthy habits. Counseling, positive reinforcement, and motivation could help patients avoid weight regain [61].

Our study demonstrates that there is not a single strategy for body weight management; in fact, different nutritional protocols (even if in different time spans) can reach the same result in terms of both anthropometrical parameters and body composition changes. It is still necessary to understand patients’ needs and health status to define a “tailor-made” nutritional treatment.

## Figures and Tables

**Figure 1 ijerph-19-13040-f001:**
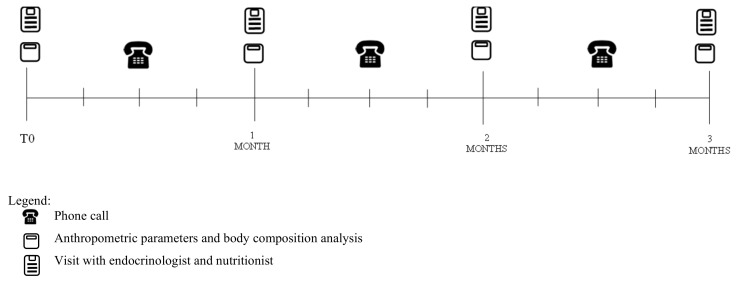
Study protocol.

**Figure 2 ijerph-19-13040-f002:**
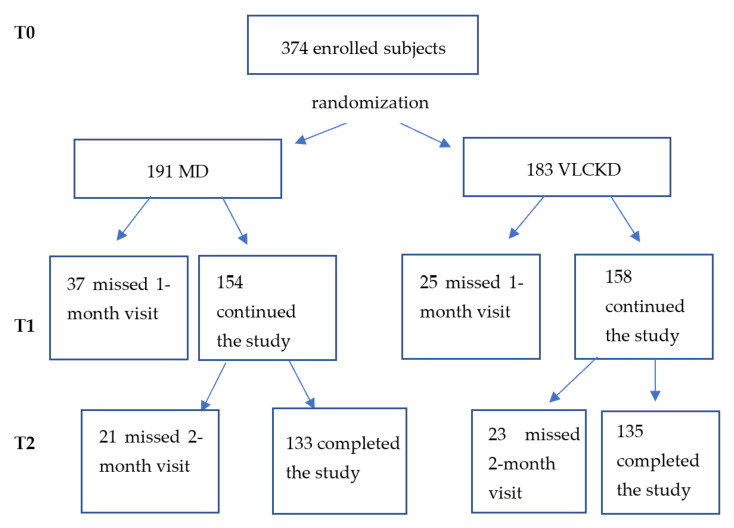
Flow—chart of the enrollment phase.

**Table 1 ijerph-19-13040-t001:** Anthropometric and Body composition changes in MD and VLCKD groups (mean ± SD).

	MD Group (*n* = 133)		Trend of Variation	*p*-Value	VLCKD Group (*n* = 135)		Trend of Variation	*p*-Value	*p*–Value Intergroup
	T0	T3M			T0	T1M			
**Weight (kg)**									
General population	88.61 ± 15.26	81.86 ± 14.54	↓	**<0.0001**	91.81 ± 18.45	85.17 ± 17.07	↓	**<0.0001**	0.6604
Male	98.67 ± 14.05	91.08 ± 13.43	↓	**<0.0001**	107.99 ± 19.5	99.56 ± 18.07	↓	**<0.0001**	0.1658
Female	84.73 ± 13.93	78.30 ± 13.40	↓	**<0.0001**	86.36 ± 14.54	80.33 ± 13.73	↓	**<0.0001**	0.1316
**Body Mass Index (BMI; kg/m^2^)**									
General population	32.14 ± 4.68	29.67 ± 4.53	↓	**<0.0001**	33.54 ± 5.49	31.14 ± 5.15	↓	**<0.0001**	0.4831
Male	32.44 ±3.8	29.94 ± 3.81	↓	**<0.0001**	35.05 ± 5.81	32.33 ± 5.41	↓	**<0.0001**	0.2778
Female	32.03 ± 4.99	29.57 ± 4.8	↓	**<0.0001**	33.03 ± 5.30	30.74 ± 5.02	↓	**<0.0001**	0.1857
**Waist Circumference (cm)**									
General population	99.78 ± 14.14	92.92 ± 13.47	↓	**<0.0001**	99.18 ± 15.46	93.44 ± 14.68	↓	**<0.0001**	**0.0010**
Male	106.86 ± 12.71	99.85 ± 12.73	↓	**<0.0001**	112.69 ± 14.28	106.33 ± 13.53	↓	**<0.0001**	0.2760
Female	97.06 ± 13.78	90.26 ± 12.84	↓	**<0.0001**	94.63 ± 13.04	89.09 ± 12.36	↓	**<0.0001**	**0.0020**
**Phase angle (°)**									
General population	6.23 ± 0.81	6.17 ±0.8	__	0.0812	6.41 ± 0.66	6.40 ± 0.73	__	0.3067	0.3324
Male	6.84 ± 0.86	6.8 ± 0.84	__	0.5593	6.88 ± 0.50	6.84 ± 0.47	__	0.4831	0.9775
Female	5.99 ± 0.66	5.93 ± 0.64	__	0.0873	6.25 ± 0.63	6.26 ± 0.74	__	0.3182	0.3397
**Total Body Water (%)**									
General population	46.77 ± 5.46	49.04 ± 6.07	↑	**<0.0001**	45.53 ± 5.32	47.11 ± 5.66	↑	**<0.0001**	**0.0017**
Male	52.36 ± 4.76	54.27 ± 5.44	↑	**<0.0001**	51.12 ± 4.24	52.27 ± 4.60	↑	**0.0012**	0.0676
Female	44.61 ± 4.00	47.02 ± 5.03	↑	**<0.0001**	43.65 ± 4.21	45.37 ± 4.88	↑	**<0.0001**	**0.0076**
**Extracellular Water (%)**									
General population	44.66 ± 3.7	44.82 ± 3.75	__	0.3005	42.28 ± 7.29	41.96 ± 6.88	__	0.1292	0.0659
Male	42.78 ± 4.38	42.85 ± 4.21	__	0.7646	42.43 ± 3.43	41.95 ± 3.48	__	0.0912	0.1342
Female	45.38 ± 3.13	45.58 ± 3.27	__	0.3189	42.23 ± 8.20	41.97 ± 7.71	__	0.3190	0.1656
**Intracellular Water (%)**									
General population	54.69 ± 5.45	54.50 ± 5.48	__	0.2410	53.86 ± 9.03	54.19 ± 9.06	__	0.3268	0.1655
Male	54.79 ± 9.3	54.74 ± 9.16	__	0.8481	57.57 ± 3.43	57.25 ± 4.58	__	0.6066	0.7454
Female	54.65 ± 3.15	54.41 ± 3.15	__	0.2142	52.62 ± 9.95	53.16 ± 9.94	__	0.1707	0.0797
**Fat-Free Mass (%)**									
General population	63.8 ± 7.4	66.68 ± 8.23	↑	**<0.0001**	61.92 ± 6.97	64.15 ± 7.45	↑	**<0.0001**	**0.0373**
Male	71.15 ± 6.4	73.58 ± 7.61	↑	**0.0003**	69.09 ± 5.27	71.07 ± 5.31	↑	**<0.0001**	0.5024
Female	60.96 ± 5.6	64.03 ± 6.82	↑	**<0.0001**	59.51 ± 5.71	61.82 ± 6.59	↑	**<0.0001**	**0.0251**
**Fat Mass (%)**									
General population	36.16 ± 7.41	33.01 ± 8.26	↓	**<0.0001**	38.29 ± 7.46	36.12 ± 8.12	↓	**<0.0001**	**0.0006**
Male	28.77 ± 6.31	25.9 ± 7.11	↓	**<0.0001**	30.89 ± 5.26	28.93 ± 5.31	↓	**<0.0001**	0.0689
Female	39.01 ±5.63	35.75 ± 6.97	↓	**<0.0001**	40.78 ± 6.37	38.54 ± 7.45	↓	**<0.0001**	**0.0034**
**Body Cellular Mass (%)**									
General population	54.67 ± 4.17	54.12 ± 4.67	__	0.0547	53.87 ± 7.54	53.90 ± 7.83	__	0.8232	0.0756
Male	57.5 ± 3.8	57.21 ± 3.61	__	0.3037	57.24 ± 2.28	56.88 ± 2.92	__	0.2677	0.8884
Female	53.57 ± 3.79	52.93 ± 4.49	__	0.0885	52.73 ± 8.32	52.90 ± 8.68	__	0.3838	0.0518

Legend: = ↓ decrease; = ↑ increase; __= no variation; 3 M = (3 months); 1 M = (1 month).

**Table 2 ijerph-19-13040-t002:** Anthropometric and Body composition changes in MD and VLCKD groups according to age (mean ± SD).

	MD Group (*n* = 78)		Trend of Variation	*p*-Value	VLCKD Group (*n* = 78)		Trend of Variation	*p*-Value	*p*-Value Intergroup
	T0	T3M			T0	T1M			
**Body weight (kg)**									
Under 50	88.09 ± 16.23	81.28 ± 15.42	↓	**<0.0001**	94.01 ± 19.69	87.19 ± 18.12	↓	**<0.0001**	0.9892
Over 50	89.35 ± 13.86	82.67 ± 13.29	↓	**<0.0001**	88.79 ± 16.3	82.4 ± 15.23	↓	**<0.0001**	0.4520
**Body Mass Index (BMI; kg/m^2^)**									
Under 50	31.34 ± 4.07	28.95 ± 3.93	↓	**<0.0001**	33.86 ± 5.79	31.44 ± 5.42	↓	**<0.0001**	0.8271
Over 50	33.29 ± 5.27	30.71 ± 5.14	↓	**<0.0001**	33.1 ± 5.05	30.73 ± 4.77	↓	**<0.0001**	0.2167
**Waist circumference (cm)**									
Under 50	96.54 ± 13.38	89.58 ± 12.48	↓	**<0.0001**	98.53 ± 17.32	92.78 ±16.47	↓	**<0.0001**	**0.0076**
Over 50	104.39 ± 14.04	97.67 ± 13.51	↓	**<0.0001**	100.07 ± 12.58	94.34 ±11.89	↓	**<0.0001**	0.0577
**Phase angle (°)**									
Under 50	6.43 ± 0.74	6.36 ± 0.74	__	0.0949	6.55 ± 0.62	6.53 ±0.72	__	0.3021	0.4144
Over 50	5.95 ± 0.84	5.90 ± 0.80	__	0.4307	6.22 ± 0.67	6.23 ± 0.71	__	0.8538	0.6885
**Total Body Water (%)**									
Under 50	47.26 ± 4.87	49.65 ± 5.53	↑	**<0.0001**	45.03 ± 5.23	46.58 ± 5.39	↑	**<0.0001**	**0.0040**
Over 50	46.06 ± 6.19	48.17 ± 6.72	↑	**<0.0001**	46.20 ±5.4	47.83 ± 5.99	↑	**<0.0001**	0.1547
**Extracellular Water (%)**									
Under 50	43.94 ± 3.42	44.15 ± 3.66	__	0.2615	40.27 ±8.49	40.21 ± 8.16	__	0.7691	0.3320
Over 50	45.68 ± 3.86	45.77 ± 3.7	__	0.7318	45.03 ± 3.82	44.36 ± 3.4	__	0.1018	0.1189
**Intracellular Water (%)**									
Under 50	55.30 ± 5.90	55.08 ± 5.73	__	0.2455	52.61 ± 11.11	52.91 ± 10.87	__	0.0595	0.0347
Over 50	53.82 ± 4.92	53.69 ± 5.06	__	0.6270	55.59 ± 4.47	55.94 ± 5.34	__	0.6459	0.5549
**Fat-Free Mass (%)**									
Under 50	64.57 ± 6.55	67.69 ± 7.42	↑	**<0.0001**	61.7 ±6.53	64.01 ± 6.72	↑	**<0.0001**	**0.0326**
Over 50	62.69 ± 8.4	65.25 ± 9.14	↑	**<0.0001**	62.24 ± 7.58	64.34 ± 8.41	↑	**<0.0001**	0.4098
**Fat Mass (%)**									
Under 50	35.36 ± 6.57	32.03 ± 7.46	↓	**<0.0001**	38.77 ± 7.33	36.62 ± 7.91	↓	**<0.0001**	**0.0019**
Over 50	37.3 ±8.40	34.4 ± 9.19	↓	**<0.0001**	37.64 ± 7.66	35.44 ±8.41	↓	**<0.0001**	0.1122
**Body Cellular Mass (%)**									
Under 50	55.72 ± 3.88	54.94 ± 5.04	__	0.0760	53.14 ± 8.86	53.02 ± 9.04	__	0.5689	0.1667
Over 50	53.17 ± 4.14	52.95 ± 3.83	__	0.4667	54.86 ±5.13	55.11 ± 5.65	__	0.3944	0.2631

Legend: = ↓ decrease; = ↑ increase; __= no variation; 3 M = (3 months); 1 M = (1 month).

**Table 3 ijerph-19-13040-t003:** Anthropometric and body composition changes in MD and VLCKD groups according to BMI (mean ± SD).

	MD Group (*n* = 46)		Trend of Variation	*p*-Value	VLCKD Group (*n* = 38)		Trend of Variation	*p*-Value	*p*-Value Intergroup
	T0	T3M			T0	T1M			
**Body Weight (kg)**									
Subjects with overweight	75.21 ± 8.58	69.06 ± 8.21	↓	**<0.0001**	75.76 ± 6.94	70.26 ± 6.49	↓	**<0.0001**	0.1254
Subjects with obesity	95.7 ± 13.11	88.62 ± 12.47	↓	**<0.0001**	98.09 ± 17.74	91.01 ± 16.37	↓	**<0.0001**	0.9878
**Body Mass Index (BMI; kg/m^2^)**									
Subjects with overweight	27.55 ± 1.56	25.35 ± 1.73	↓	**<0.0001**	27.67 ± 1.37	25.70 ± 1.31	↓	**<0.0001**	0.1740
Subjects with obesity	34.57 ± 3.89	31.96 ±3.83	↓	**<0.0001**	35.84 ± 4.72	33.27 ± 4.48	↓	**<0.0001**	0.7133
**Waist Circumference (cm)**									
Subjects with overweight	88.74 ± 9.10	81.83 ± 8.52	↓	**<0.0001**	87.06 ± 11.72	82.05 ± 11.11	↓	**<0.0001**	**0.0020**
Subjects with obesity	105.77 ± 12.62	98.79 ± 11.83	↓	**<0.0001**	103.93 ± 14.13	97.89 ± 13.49	↓	**<0.0001**	**0.0289**
**Phase angle (°)**									
Subjects with overweight	6.29 ± 0.70	6.21 ± 0.67	__	0.0812	6.30 ± 0.63	6.21 ± 0.66	__	0.0818	0.9077
Subjects with obesity	6.2 ± 0.87	6.15 ± 0.86	__	0.2534	6.45 ± 0.67	6.48 ± 0.74	__	0.5072	0.9390
**Total Body Water (%)**									
Subjects with overweight	49.74 ± 4.87	52.53 ± 5.49	↑	**<0.0001**	48.24 ± 3.89	50.23 ± 3.57	↑	**<0.0001**	**0.0340**
Subjects with obesity	45.19 ± 5.12	47.2 ± 5.55	↑	**<0.0001**	44.46 ± 5.44	45.88 ± 5.87	↑	**<0.0001**	**0.0301**
**Extracellular Water (%)**									
Subjects with overweight	55.82 ± 3.16	55.46 ± 2.93	__	0.3005	42.64 ± 7.75	42.53 ± 7.26	__	0.8379	0.6403
Subjects with obesity	44.87 ± 3.95	45.03 ± 4.10	__	0.4031	42.14 ± 7.13	41.74 ± 6.75	__	**0.0423**	**0.0409**
**Intracellular Water (%)**									
Subjects with overweight	44.26 ± 3.15	44.41 ± 2.97	__	0.2410	54.01 ± 9.74	53.22 ± 8.43	__	0.1883	0.4870
Subjects with obesity	54.09 ± 6.4	53.99 ± 6.4	__	0.6229	53.81 ± 8.78	54.57 ± 9.30	__	0.0547	0.0589
**Fat-Free Mass (%)**									
Subjects with overweight	67.96 ± 6.57	71.13 ± 7.57	↑	**<0.0001**	65.58 ± 6.59	68.16 ± 6.80	↑	**<0.0001**	0.4078
Subjects with obesity	61.59 ± 6.87	64.33 ± 7.61	↑	**<0.0001**	60.49 ±6.61	62.58 ± 7.13	↑	**<0.0001**	0.0516
**Fat Mass (%)**									
Subjects with overweight	32.03 ± 6.57	28.2 ± 7.32	↓	**<0.0001**	34.32 ± 6.59	31.84 ± 6.80	↓	**<0.0001**	**0.0126**
Subjects with obesity	38.35 ± 6.92	35.55 ± 7.61	↓	**<0.0001**	39.81 ± 7.26	37.8 ± 8.00	↓	**<0.0001**	**0.0237**
**Body Cellular Mass (%)**									
Subjects with overweight	54.94 ± 3.39	53.81 ± 5.47	__	0.0547	54.28 ± 8.61	53.91 ± 8.86	__	0.1665	0.3538
Subjects with obesity	54.52 ± 4.54	54.28 ± 4.21	__	0.2471	53.7 ± 7.12	53.9 ± 7.44	__	0.3534	0.1436

Legend: ↓ decrease; ↑ increase; __ no variation; 3 M (3 months); 1 M (1 month).

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
