# Peer review of "Mediterranean Diet versus Very Low-Calorie Ketogenic Diet: Effects of Reaching 5% Body Weight Loss on Body Composition in Subjects with Overweight and with Obesity—A Cohort Study"

_ijerph, 2022, doi:10.3390/ijerph192013040_

Round 1

Reviewer 1 Report

Abstract:

….(p<0.0001) and FM reduction (p<0.0001) but MD determined a higher reduction of both -    25

FM what is it? not written in full.

 Introduction

… sity [4]. Overweight and obesity are classified based on the Body Mass Index (BMI) - 41

(BMI) reference is missing

…they were randomized to either MD or VLCKD. Participants received nutritional coun-81

Randomly? in what way?

Materials and Methods

….they were randomized to either MD or VLCKD. Participants received nutritional coun-81

Randomly in what way?

…At T0 body weight (BW) and height were recorded using a Seca 200 scale with incorpo-93

rated stadiometer.

Under what conditions?

Anthropometric parameters

…BMI was calculated as weight (kg)/ height (m2) and waist circumference (WC) was - 96

What reference values are used? WHO? Must reference!

Mediterranean Diet

….(TEE) was calculated and then a caloric restriction of 500 kcal was applied. The macro-113

On average, what was the VCT provided?

…and to reduce red meat, eggs, and dairy products to only once a week. They were in- 118

And was the consumption of olive oil recommended?

Very low - calorie Ketogenic Diet

…Multivitamin, minerals and omega 3 fatty acids were prescribed. 129

What is the composition of multivitamins?

Author Response

Dear Reviewer,

Thank you very much for your comments. Enclosed you can find the answers to your comments and you can check the corrections in the text. The manuscript was sent to MDPI to undergo English editing. The extensive English Editing is only in the final version of the manuscript, not in the version with corrections.     

Reviewer 2 Report

This study includes a valuable purpose for obesity epidemiology, and the results may contribute to the development of weight control and public health.

However, there are serious problems, the manuscript needs to make a significant correction for the reviewing and proofreading of it. In particular, the following points should be revised.

1. Please indicate the study design in the title.

2. Please provide a flowchart of the exclusion criteria for the study subjects.

3. Please indicate the characteristics of each group (Mediterranean diet vs Very Low-Calorie Ketogenic diet) by gender (preferably in the first table).

4 Were height measurements also repeated in this study? If so, was there any change in height? Probably no change, but needs to be clearly indicated.

5. Please indicate the rationale for separating the ages at 50 years (e.g., references).

6. After revising the above points, check the STROBE guideline for cohort study. Please submit the checklist. (https://www.strobe-statement.org/checklists/).

Author Response

Dear Reviewer,

Thank you very much for your comments. Below you can find the answers to your comments and you can check the corrections in the text. The manuscript was sent to MDPI to undergo English editing. The extensive English Editing is only in the final version of the manuscript, not in the version with corrections.   

Reviewer 3 Report

The manuscript titled Mediterranean diet vs Very Low-Calorie Ketogenic diet: Effects of reaching 5% body weight loss on body composition in subjects with overweight and obesity was reviewed. Although the topic of the manuscript is original, the academic writing style is not enough/The manuscript is not well-written and all sections including the introduction, method, results, and discussion should be improved. The authors should discuss their results with more up-to-date sources and in a more critical and rigorous manner. Please find some suggestions in the attached file.

Author Response

(The authors gave the same response as above.)

Round 2

Reviewer 2 Report

The authors have appropriately revised and explained manuscript toward reviewer's comments.

Author Response

Dear Reviewer,

Thank you very much for your comments. We are pleased that our corrections are in line with your previous requests.

Regarding the English editing, we had already sent the manuscript to MDPI to extensive English editing.

Reviewer 3 Report

-

Author Response

Dear Reviewer,
thanks for your comments. We have submitted the manuscript version with all comments and corrections requested in round 1. In round 2 we saw a table, but as it is the same as round 1, we think that maybe you couldn’t visualize our corrections as we had submitted a final version where corrections and comments were not visible. 
